# Microstructure Optimization and Coercivity Enhancement of Sintered NdFeB Magnet by Grain Boundary Diffusion of Multicomponent Tb_60_Pr_10_Cu_10_Al_10_Zn_10_ Films

**DOI:** 10.3390/ma16083131

**Published:** 2023-04-16

**Authors:** Jingbin Huang, Min Huang, Fang Wang, Zhanyong Wang, Jian Zhang

**Affiliations:** 1School of Materials Science and Engineering, Shanghai Institute of Technology, Shanghai 201418, China; huangjingbin@nimte.ac.cn; 2CAS Key Laboratory of Magnetic Materials and Devices, Ningbo Institute of Materials Technology and Engineering, Chinese Academy of Sciences, Ningbo 315201, China; huangmin@nimte.ac.cn; 3Zhejiang Province Key Laboratory of Magnetic Materials and Application Technology, Ningbo Institute of Materials Technology and Engineering, Chinese Academy of Sciences, Ningbo 315201, China; 4School of Materials and Chemical Engineering, Ningbo University of Technology, Ningbo 315211, China; wangfangch@nbut.edu.cn

**Keywords:** grain boundary diffusion, multicomponent film, coercivity, microstructural optimization

## Abstract

The use of magnetron sputtering film as a diffusion source was recently achieved in the industrial production of important grain-boundary-diffusion magnets. In this paper, the multicomponent diffusion source film is explored to optimize the microstructure of NdFeB magnets and improve their magnetic properties. Multicomponent Tb_60_Pr_10_Cu_10_Al_10_Zn_10_ films of 10 μm in thickness and single Tb films of 10 μm in thickness were deposited on commercial NdFeB magnets’ surfaces by magnetron sputtering as diffusion sources for grain boundary diffusion. The effects of diffusion on the microstructure and magnetic properties of the magnets were investigated. The coercivity of multicomponent diffusion magnets and single Tb diffusion magnets increased from 11.54 kOe to 18.89 kOe and 17.80 kOe, respectively. The microstructure and element distribution of diffusion magnets were characterized by scanning electron microscope and transmission electron microscopy. The multicomponent diffusion facilitates the infiltration of Tb along grain boundaries, rather than entering the main phase, thereby improving the Tb diffusion utilization. Furthermore, compared to the Tb diffusion magnet, the thicker thin-grain boundary was observed in multicomponent diffusion magnets. This thicker thin-grain boundary can effectively serve as the impetus for the magnetic exchange/coupling between grains. Therefore, the multicomponent diffusion magnets have higher coercivity and remanence. The multicomponent diffusion source has an increased mixing entropy and decreased Gibbs free energy, and it therefore does not easily enter the main phase but is retained in the grain boundary, thus optimizing the microstructure of the diffusion magnet. Our results show that the multicomponent diffusion source is an effective route for fabricating diffusion magnets with high performance.

## 1. Introduction

The main phase Nd_2_Fe_14_B with a complex tetragonal structure has a strong anisotropy field and high saturation magnetization (μ_0_H_c_ = 7.5 T, μ_0_M_s_ = 1.6 T) [1,2]. As a material medium for energy conversion, NdFeB magnets play a critical role in promoting green energy today [3,4]. They have been widely used in various important fields, such as electric vehicles, power generation, medical, maglev trains, etc. By adjusting the composition and process, the remanence (B_r_) and maximum energy product ((BH)_max_) of NdFeB magnets have reached their theoretical values closely. However, the coercivity of sintered NdFeB permanent magnets is only about 30% of their theoretical value [5]. It is of great practical significance to further develop and innovate NdFeB permanent magnets with higher coercivity [6].

The grain boundary diffusion (GBD) technology with heavy rare earth (HRE) elements (Dy/Tb) infiltration is considered to be the most effective at improving the coercivity of NdFeB magnets without excessive consumption of HRE [7]. Tb_2_Fe_14_B and Dy_2_Fe_14_B phases have a magnetocrystalline anisotropy field that is much higher than that of Nd_2_Fe_14_B, which can effectively suppress the reverse magnetization process [8]. The diffusion source rich in Dy/Tb can diffuse into the magnet interior through the grain boundary (GB), and Nd atoms are replaced to form a rich-Dy/Tb shell on the outer layer of the Nd_2_Fe_14_B grains. The improvement of coercivity is attributed to the magnetic hardening effect of the Tb/Dy-rich shell [7]. In addition, due to the antiferromagnetic coupling between Tb/Dy and Fe [9], a single Tb or Dy diffusion source can make the remanence decline [10]. Although increasing the Tb/Dy content in the diffusion source can form a high diffusion gradient, which is helpful for the formation of a Dy/Tb-rich shell structure, it also directly leads to the low utilization of HRE elements [11,12]. The coercivity of diffused magnets is restricted by the thickness of the Dy/Tb core–shell structure and HRE diffusion depth [13,14]. The diffusion sources such as PrTbAl [15], PrTbCuAl [16], TbAl [17], DyZn [18], DyMg [19] and PrCoAl [20] have been usually utilized in diffusion magnets. These diffusion sources are beneficial for improving the microstructure of magnets, which is responsible for their high coercivity. It is found that these grain boundary diffusions promote the formation of RE-rich core–shell structure and also continuous thin grain boundaries. CuAl element can improve the wettability of the grain boundary phase [17,21,22].

Varieties of routes have been developed to coat HRE sources on Nd-Fe-B magnets, such as dipping deposition [23,24], electrophoretic deposition [11,25], magnetron sputtering [26,27], etc. Among them, magnetron sputtering film technology is the most important method and has been recently achieved in the industrial production of grain boundary diffusion magnets utilized in new energy vehicles. In this work, five-element Tb_60_Pr_10_Cu_10_Al_10_Zn_10_ film and single Tb film were prepared by magnetron sputtering as diffusion sources of commercial NdFeB magnets. We propose that the multielement sources diffusing along grain boundaries can increase the mixed entropy of grain boundary phases and reduce their Gibbs free energy, thereby facilitating the diffusion of heavy rare earth Tb along grain boundaries.

## 2. Materials and Methods

The commercial N52 sintered Nd-Fe-B magnets were cut into the size of 25 mm × 3 mm × 2.5 mm (The c-axis is parallel to the 2.5 mm). The nominal composition of Nd_24.4_Pr_6.1_Fe_68_B_1_X_0.5_ (wt.%, X = Al, Cu, Nb, Ga) was analyzed by the ICP method. The cutting magnet was ground with silicon carbide sandpaper. After acetone cleaning and vacuum drying, the experimental magnets were obtained. Tb_60_Pr_10_Cu_10_Al_10_Zn_10_ (Tb60) films of 10 μm in thickness or Tb (Tb) films of 10 μm in thickness were deposited on one side of the magnet (25 mm × 3 mm surface) by magnetron sputtering system. The commercial Tb/Pr (99.9%) and Cu/Al/Zn (99.995%) targets were used. The vacuum in the sputtering chamber is better than 3 × 10^−6^ Pa. During deposition, Ar pressure for sputtering was maintained at 0.8 Pa and Ar flow was 32 sccm. The sputtering power for Tb/Pr/Cu/Al/Zn target was 120 W/24 W/10 W/83 W/13 W, respectively. Then the pre-diffused sample was sealed in a quartz tube (<3 × 10^−4^ Pa). Finally, the sealed samples in a muffle furnace were heat treated at 800 °C for 8 h and annealed at 500 °C for 2 h. The magnetic properties were tested at room temperature by magnetic property measurement system (MPMS, Quantum Design, San Diego, CA, USA) equipped with a 7 T vibrating sample magnetometer (VSM). The phase and crystal structure before and after diffusion were studied using an X-ray diffractometer (XRD-D8 ADVANCE, Bruker, Karlsruhe, Germany). The microstructure and elemental distribution of diffusion magnets were observed by scanning electron microscope (SEM, FEI Quanta FEG 250, FEI, Oregon City, OR, USA) with an energy-dispersive X-ray spectrometer (EDS). Talos F200X scanning/transmission electron microscopy (TEM, ThemoFisher, Waltham, MA, USA) was exploited to analyze the microstructure of the grain boundary (GB) phase.

## 3. Results and Discussion

The demagnetization curves and properties of the original and diffusion magnets are shown in Figure 1 and Table 1. After Tb60 and Tb films diffusion, the H_cj_, B_r_ and (BH)_max_ were 18.89 kOe/17.80 kOe, 14.39 kGs/13.50 kGs and 378.94 kJ m^−3^/316.49 kJ m^−3^, respectively. Compared with the original magnet, the coercivity increased by 63.47% and 54.78%, respectively. Meanwhile, the coercivity increment of the Tb60 diffusion magnet is 7.35 kOe, which is higher than the 6.26 kOe of the Tb diffusion magnet. It is worth noting that the H_cj_, B_r_ and (BH)_max_ of the Tb60 diffusion magnet are all larger than that of the Tb diffusion magnet, thus showing the great superiority of the multicomponent diffusion.

X-ray diffraction patterns of the original and the Tb60 and Tb diffusion magnets are shown in Figure 2. The diffraction peaks match (Nd, Pr)_2_Fe_14_B phases (Wyckoff positions of space group, P42/mnm (136); JCPDS, 89-3632) and RE-rich phases (Wyckoff positions of space group, P63/mmc (194); JCPDS, 65-3424) well. However, the 2:14:1 matrix phase peak shifts slightly towards the higher angles after grain boundary diffusion. According to the Bragg equation, 2dsinθ = nλ [28], where d is the crystal plane spacing, θ is the angle between the incident X-ray and the crystal plane, and λ represents the X-ray wavelength (using Cu-Kα radiation, λ = 0.154056 nm), a high angle shift means that the crystal plane spacing and lattice parameters were reduced [29,30,31]. Actually, the Tb atom radius is smaller than that of the Nd/Pr atom. It is considered that the infiltration of the Tb element causes partial Nd/Pr atoms of the (Nd, Pr)_2_Fe_14_B main phase to be replaced [15]. As a result, the lattice constant decreases. It is emphasized that the peak shift of the Tb60 diffusion magnet is similar to that of the Tb diffusion magnet, indicating that the Tb substitution total amount of the two diffusion magnets is approximate in the near surface.

Figure 3 depicts the backscattered electron images of diffusion magnets within the depth range of 105 μm. Bright white triple grain boundaries and black Nd_2_Fe_14_B grains wrapped in gray Tb-rich shells are clearly visible in both magnets. The difference is that the gray Tb-rich shell occupies almost the entire main phase grain at the surface of Tb diffusion magnet (Figure 3b), forming a Tb accumulation zone [32] (about 30 μm). With the increasing diffusion depth, the thickness of the gray Tb-rich shell gradually decreased, and the Tb-rich layer could still be observed at a diffusion depth of 75 μm. In a Tb60 magnet (Figure 3a), its shell thickness is thinner than that of the Tb diffusion magnet at the same diffusion depth, and the Tb-rich shell is visible within a diffusion depth of 45 μm. Obviously, the Tb60 diffusion magnet has a shallower and thinner Tb-rich shell structure, indicating that most of the Tb is retained in the grain boundary rather than entering the main phase to form a thick Tb-rich shell during multicomponent diffusion. Usually, in heavy rare earth grain boundary diffusion magnets, the wider the distribution of the core–shell structure, the more beneficial it is for the improvement of coercivity [14,15]. However, in our experiments, the distribution depth of the core–shell structure in the Tb diffusion magnet is superior to that of the Tb60 diffusion magnet, but its coercivity is lower. This indicates that the core–shell microstructure and its depth have a limited effect on the coercivity [11]. Other factors affecting coercivity may play a more important role.

Figure 4 shows the elemental distributions in the diffusion magnets within the depths of 50 μm. The presence of a Tb-rich shell on the surface of the main phase grain is further confirmed in both magnets. In the Tb diffusion magnet (Figure 4b), the Nd/Pr atoms are extruded out and enriched at the triple grain boundary. The Tb atoms diffuse almost into the entire grain, and thicker Tb-rich shells are still observed far away from the magnet surface. For Tb60 diffusion magnets, only very thin Tb-rich shells are observed at the surface (Figure 4a). This suggests that, for multicomponent diffusion magnets, the Tb element does not easily enter the main phase, and most of the Tb is retained in the grain boundary or at the surface. It was also observed that Nd/Pr/Cu/Al/Zn atoms were enriched at the triple grain boundary. Due to the low resolution of SEM, thin grain boundaries in the magnets cannot be seen clearly. We can speculate that these elements are also concentrated in thin grain boundaries. It is worth mentioning that the gathering of Cu and Zn elements reduces the melting point of the grain boundary phase [16,20], and this helps improve the wettability of the grain boundary phase [33].

The variation of the Tb concentration in the grain boundary of the diffusion magnet is shown in Figure 5. After the magnet is annealed at a high temperature, the multicomponent diffuses into the magnet along the liquid grain boundary. The diffusion magnet used in this study has only one face perpendicular to the NdFeB magnet c-axis as the diffusion surface; thus, it can be simplified to a diffusion model whose components are not affected by diffusion at one end. The Grube solution for one-dimensional diffusion under constant source conditions according to Fick’s second law is as follows [34]:
(1)$$c\;(x,\;t)\;=\;c_{1}-(c_{1}-c_{0})\;\mathrm{erf}\;(\frac{x}{\sqrt{Dt}}),$$
where c (x, t) represents the element concentration as a function of diffusion depth (x) and time (t), c_1_ represents the concentration of diffused elements at the surface of the magnet, c_0_ represents the initial concentration of diffusion elements at infinity from the magnet and D is the diffusion coefficient. The data of Tb content change when the depth in the grain boundary of the diffusion magnet is fitted to the equation to obtain the fitting curve and diffusion coefficient. As the diffusion depth increases, the Tb concentration in the Tb60 diffusion magnet decreases more slowly than that in the Tb diffusion magnet. The diffusion coefficient of Tb60 diffusion magnets is just slightly lower than that of Tb diffusion magnets. Although the Tb content in the Tb60 diffusion source is significantly lower, the Tb atoms’ diffusion depth is not much different from that of the Tb diffusion magnet, thus indicating that most of Tb in the Tb60 diffusion magnet is retained in the grain boundary or grain surface and does not enter the main phase grain. This reflects the advantages of multicomponent diffusion. In the Tb60 diffusion magnet, the multicomponent elements diffuse into the magnet along the grain boundary, reducing the melting point of the grain boundary phase. More importantly, the multicomponent increases the mixed entropy of the grain boundary phase and reduces its Gibbs free energy, causing most of the Tb to be retained in the grain boundary or grain surface layer without entering the main phase grain too much, and therefore inhibiting the increase of the thickness of the Tb-rich shell. This is also the reason why the core–shell structure of the Tb60 diffusion magnet is shallow. The addition of multicomponent greatly improves the utilization efficiency of Tb.

The near-surface microstructure of the diffusion magnets is also characterized by HRTEM. The Tb60 diffusion magnet (Figure 6a) has thicker (about 11 nm) and more continuous thin grain boundary phases. However, a thinner and discontinuous interface phase is observed in the Tb diffusion magnet (Figure 6b). The reason for the thickening of grain boundaries after multicomponent diffusion can be explained by the Gibbs free energy. For multi-alloys, G_multi-elements_ = H_multi-elements_ − TS_multi-elements_. The multicomponent diffuses into the magnet, causing an increased entropy and decreased Gibbs free energy of the grain boundary phase. The stability of the multicomponent liquid grain boundary phase is increased, so that most of the diffusion elements are retained in the grain boundary and do not easily enter the main phase, thereby broadening the thin grain boundary in the magnet and increasing its continuity. The thicker and continuous thin grain boundaries can effectively inhibit the magnetic exchange coupling between grains [35,36]. This is why Tb60 diffusion magnets still have higher coercivity than Tb diffusion magnets when only 60% Tb content is contained in the diffusion source. Therefore, multicomponent grain boundary diffusion has greater advantages in optimizing the microstructure and improving the magnetic properties.

Figure 7 shows the temperature dependence of the coercivities for Tb and Tb60 diffusion magnets. The coercivity temperature coefficient (β_Hcj_) is used to describe the stability of magnets at high temperatures. The β_Hcj_ is calculated by the following formula:β_Hcj_ = [H_cj_(T) − H_cj_(T_0_)]/[H_cj_(T)(T − T_0_)] × 100%,(2)
where T_0_ is the contrast temperature, T is the highest temperature, H_cj_(T_0_) is the coercivity of a magnet at contrast temperature and H_cj_(T) is the coercivity of a magnet at the highest temperature. The original magnet, Tb60 diffusion magnet and Tb diffusion magnet β_Hcj_ are −0.619%/K, −0.553%/K and −0.635%/K, respectively. The β_Hcj_ value of the Tb60 diffusion magnet is increased from −0.619%/K to −0.553%/K, while the β_Hcj_ value of single Tb diffusion magnet is decreased from −0.619%/K to −0.635%/K. The │β_Hcj_│ value of the Tb60 diffusion magnet was lower than that of the original magnets, indicating that the thermal stability of the Tb60 diffusion magnet was improved [37], which may be related to the optimization of the boundary phase after multicomponent diffusion. On the contrary, the single Tb diffusion magnet has poor thermal stability compared to the original magnet.

## 4. Conclusions

After the boundary diffusion by the multicomponent Tb_60_Pr_10_Cu_10_Al_10_Zn_10_ film and single Tb film, the magnet coercivity increased from 11.54 kOe to 18.89 kOe and 17.80 kOe, respectively. Although the multicomponent diffusion source contains only 60 at. % Tb, the coercivity of the diffusion magnet was improved more than that of the Tb diffusion magnet. The microstructure analysis confirms that a thinner Tb-rich shell structure and a thicker continuous thin grain boundary are formed in the Tb60 diffusion magnet. Moreover, the Tb diffusion depth of Tb60 diffusion magnet is close to that of Tb diffusion magnet due to the addition of multicomponent. The multicomponent diffusion increases the mixed entropy of the grain boundary phase and reduces its Gibbs free energy, making Tb and other diffusion elements more difficult to enter the main phase but retained in the grain boundary phase or grain surface, thus forming a thin Tb-rich shell, deep grain boundary diffusion and thickened continuous thin grain boundary phase, leading to the improvement of the coercivity. The multicomponent grain boundary diffusion has great advantages in optimizing magnet microstructure, improving Tb utilization and increasing magnetic properties. This work provides a reference for the design and development of high-performance sintered NdFeB magnets with multicomponent elements serving as diffusion sources.

## Figures and Tables

**Figure 1 materials-16-03131-f001:**
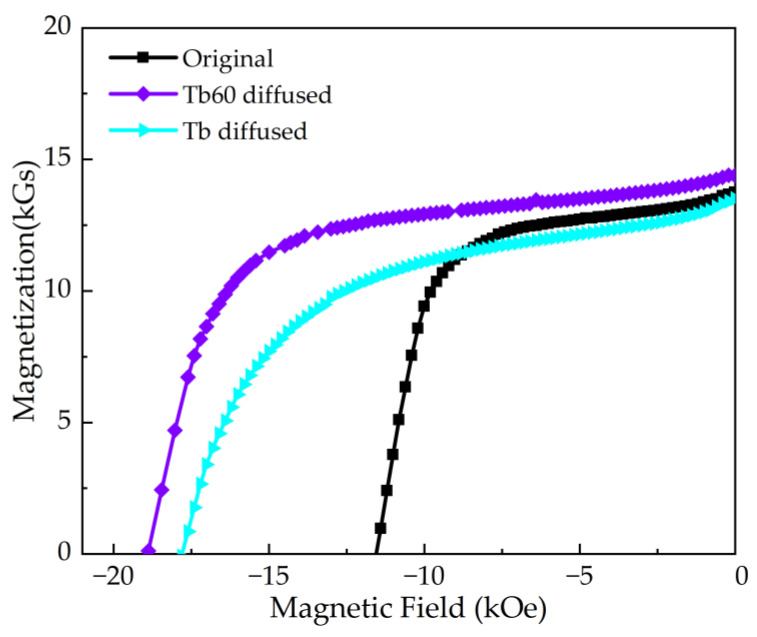
Demagnetization curves of the original and the Tb60 and Tb diffusion magnets.

**Figure 2 materials-16-03131-f002:**
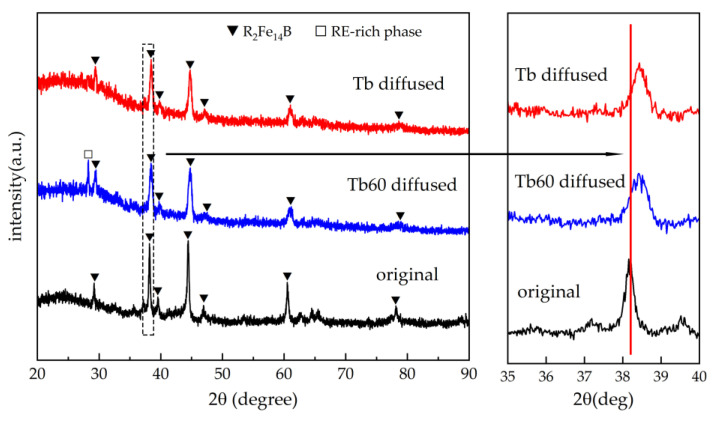
The X-ray diffraction patterns of the original and the Tb60 and Tb diffusion magnets.

**Figure 3 materials-16-03131-f003:**
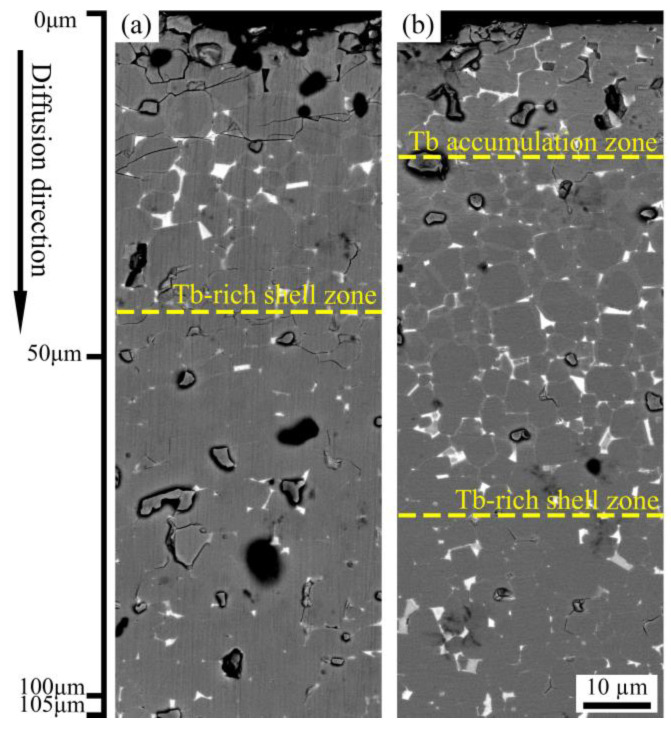
SEM images of section samples (parallel to c-axis) at depth from 0 to 105 µm: (**a**) Tb60 diffusion magnet and (**b**) Tb diffusion magnet.

**Figure 4 materials-16-03131-f004:**
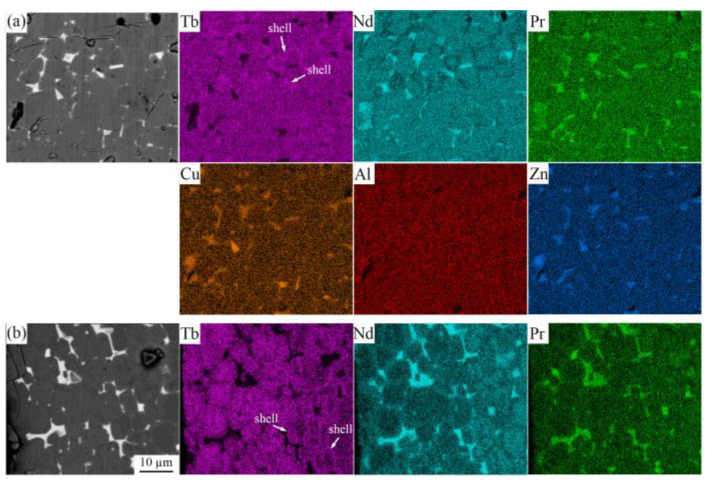
EDS mapping images on the surface of the GBD magnets at the depth of 0–50 µm: (**a**) Tb60 diffusion magnet and (**b**) Tb diffusion magnet.

**Figure 5 materials-16-03131-f005:**
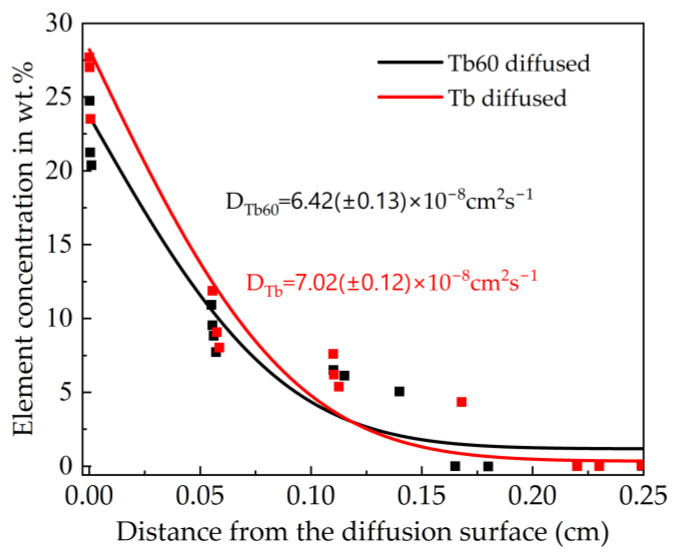
Tb concentrations at different depths and the corresponding fitting curves of Tb60 diffusion magnet and Tb diffusion magnet.

**Figure 6 materials-16-03131-f006:**
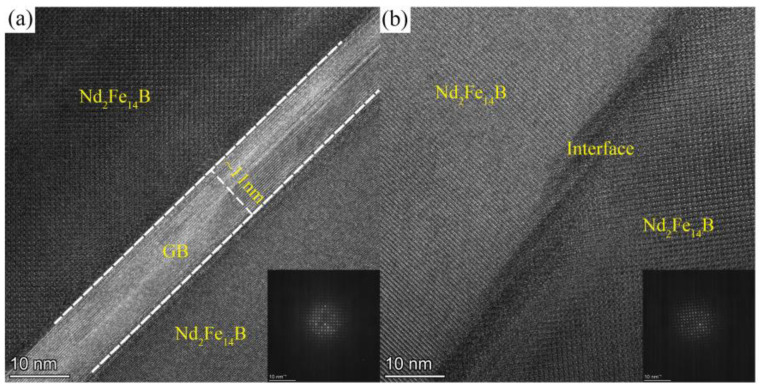
TEM images of the grain boundary structure: (**a**) Tb60 diffusion magnet and (**b**) Tb diffusion magnet.

**Figure 7 materials-16-03131-f007:**
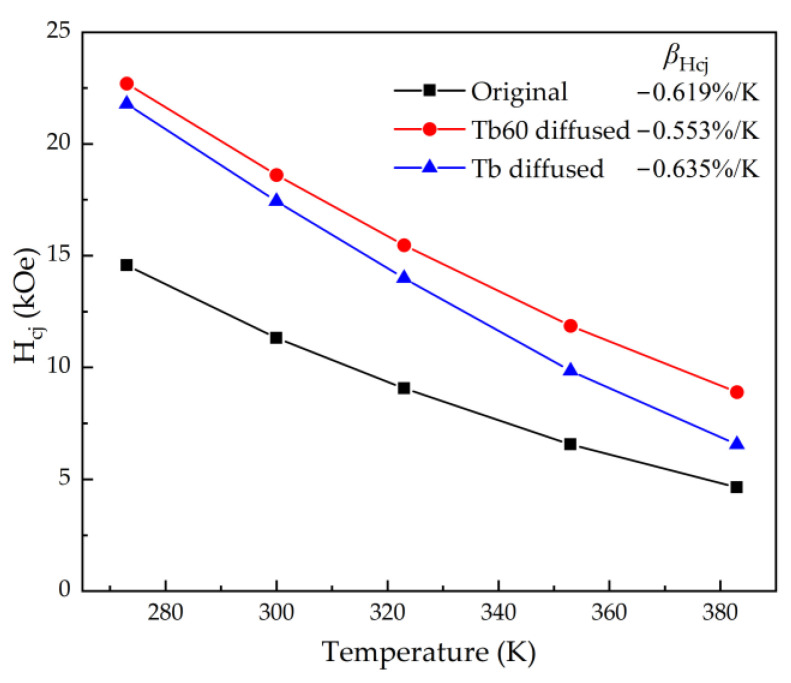
The temperature dependence of the coercivities for the original and the Tb60 and Tb diffusion magnets.

**Table 1 materials-16-03131-t001:** The magnetic properties of the original and the Tb60 and Tb diffusion magnets.

Magnet	H_cj_ (kOe)	M_r_ (kGs)	(BH)_max_ (kJ m^−3^)
Original	11.54	13.74	338.94
Tb60	18.89	14.39	378.26
Tb	17.80	13.50	316.49

## Data Availability

The data presented in this study are available upon request from the corresponding author.

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
