# Peer review of "Microstructure Optimization and Coercivity Enhancement of Sintered NdFeB Magnet by Grain Boundary Diffusion of Multicomponent Tb_60_Pr_10_Cu_10_Al_10_Zn_10_ Films"

_materials, 2023, doi:10.3390/ma16083131_

Round 1

Reviewer 1 Report

The authors described an intriguing experimental strategy for producing Tb60Pr10Cu10Al10Zn10 film and single Tb film as diffusion sources for commercial NdFeB magnets. The multicomponent diffusion source film is being investigated in order to optimize the microstructure and magnetic characteristics of NdFeB magnets. The magnet coercivity increased from 11.54kOe to 18.89kOe and 17.80kOe after boundary diffusion by the multicomponent Tb60Pr10Cu10Al10Zn10 film and single Tb film, respectively. Despite the fact that the multicomponent diffusion source includes only 60% Tb, the coercivity of the diffusion magnet is higher than that of the Tb diffusion magnet. The various experimental techniques used were sound and backed with numerous figures and supporting data. Nonetheless, there are a few small points to consider:

1-     It is preferable to write the unsteady diffusion equation (lines 150,151) separately and its parameters should be defined.

2-     Figures should be put in the text where they belong.

3-     In Figure 2 (XRD-pattern), the principal peaks broaden and their intensity diminishes, indicating that the degree of crystallinity and the crystallites size decrease with Tb additions. Is this modification affecting the magnetic characteristics of the magnets under consideration?

Author Response

Dear Reviewer,

We thank the reviewers for the efforts given in reading our manuscript (materials-2330627) and having taken the time to provide valuable comments. We have revised the manuscript according to your requirements carefully. The comments, our responses and the main modifications are listed as follows:

The authors described an intriguing experimental strategy for producing Tb60Pr10Cu10Al10Zn10 film and single Tb film as diffusion sources for commercial NdFeB magnets. The multicomponent diffusion source film is being investigated in order to optimize the microstructure and magnetic characteristics of NdFeB magnets. The magnet coercivity increased from 11.54kOe to 18.89kOe and 17.80kOe after boundary diffusion by the multicomponent Tb60Pr10Cu10Al10Zn10 film and single Tb film, respectively. Despite the fact that the multicomponent diffusion source includes only 60% Tb, the coercivity of the diffusion magnet is higher than that of the Tb diffusion magnet. The various experimental techniques used were sound and backed with numerous figures and supporting data. Nonetheless, there are a few small points to consider.

1     Comment 1: It is preferable to write the unsteady diffusion equation (lines 150,151) separately and its parameters should be defined.

Response:

Thank you for your suggestion. We modified the description of the unsteady diffusion equation in lines 149-151, and defined of parameters in detail. The sentence " The data were fitted by the unsteady-states diffusion equation c (x, t) = c1-(c1-c0) erf () [31] to obtain the diffusion coefficient" has been changed into the following section:

“After the magnet is annealed at high temperature, the multicomponent diffuses into the magnet along the liquid grain boundary. The diffusion magnet used in this study has only one face perpendicular to the Nd-Fe-B magnet c axis as the diffusion surface. So, it can be simplified to a diffusion model whose components are not affected by diffusion at one end. Grube solution for one-dimensional diffusion under constant source conditions according to Fick's second law [31]:

                   c (x, t) = c1-(c1-c0) erf ()                               (1)

where c (x, t) represents the element concentration as a function of diffusion depth (x) and time (t). c1 represents the concentration of diffused elements at the surface of the magnet. c0 represents the initial concentration of diffusion elements at infinity from the magnet. D is the diffusion coefficient. The data of Tb content change with depth in the grain boundary of the diffusion magnet are fitted to the equation to obtain the fitting curve and diffusion coefficient.”. We have revised it in the manuscript. (lines 171-184)

We also notice that the diffusion coefficient value is omitted from Figure 5 and the content of the image has been modified to add the diffusion coefficient value.

2     Comment 2: Figures should be put in the text where they belong.

Response:

We adjust Figure 1 to lines 112-113, adjust Table 1 to line 114 and bold the Hcj (kOe) and Mr (kGs) labels in Table 1, adjust Figure 2 to lines 130-131, adjust Figure 3 to lines 150-152, adjust Figure 4 to lines 167-169, adjust Figure 5 to lines 200-202.

3     Comment 3: In Figure 2 (XRD-pattern), the principal peaks broaden and their intensity diminishes, indicating that the degree of crystallinity and the crystallites size decrease with Tb additions. Is this modification affecting the magnetic characteristics of the magnets under consideration?

Response:

After the Nd-Fe-B sintered magnet is diffused by multicomponents, Tb atoms diffuse into the grain boundary and Nd2Fe14B. And the Tb atoms entering the grain will replace Nd / Pr atoms of (Nd, Pr)2Fe14B main phase to form the Tb-rich shell structure. Therefore, the surface grains of diffused magnets consist of two parts: the (Nd, Pr)2Fe14B phase in the core and the Tb-rich shell structure on the grain surface. The Tb atom radius is smaller than that of the Nd / Pr atom, so the lattice constant of the Tb-rich shell phase decreases. According to Bragg's equation, the peak position must shift to a higher angle when X-ray is scanning the Tb-rich shell. However, the centered (Nd, Pr)2Fe14B phase also interferes under X-ray scanning, but its peak does not move. So, the principal peaks broaden, which should be the result of the superposition of the two phases. This also directly proves that our diffused Tb atoms enter the grain and form a core-shell structure. The improvement of coercivity is attributed to the magnetic hardening effect of the Tb - rich shell.

Reviewer 2 Report

In the submitted manuscript entitled “Microstructure optimization and coercivity enhancement of sintered NdFeB magnet by grain boundary diffusion of multicomponent Tb60Pr10Cu10Al10Zn10 films” by J. Zhang et al to Materials/MDPI the authors have discussed sintered Nd-Fe-B magnets and optimizing their microstructure using various diffusion sources.

Nd-Fe-B magnets have been widely used in clean energy applications because of their high magnetic properties. Recently, there has been an increasing demand for Nd-Fe-B magnets with high remanence and high heat resistance for HEV traction motors. Generally, these applications require magnets with high remanence, Br, and large coercivity, Hc. Therefore, the area of research is significant and the multicomponent diffusion sources for fabricating magnets with enhanced performance have not been widely known. The paper itself is written well and seems worthwhile but the main concern of this paper and the areas that require clarification are given below.

 ·         The hot-deformed Nd-Fe-B magnets have a fine microstructure (diameter of 200–500 nm), this is one order of magnitude finer than the sintered magnets. Why have the authors chosen this approach, as the microstructural changes are lacking in this technique?

·         Hot-deformed Nd-Fe-B magnets are considered a promising candidate to achieve completely rare-earth-free Nd-Fe-B magnets with high performance. However, the authors have used sintered approach with rare-earth, please justify.

·         The originality of this work must be emphasized in the last paragraph of the introduction’ and how the coercivity exceeds than conventional magnets?

·         What is the rationale for choosing Tb60? How are the compositions optimized?

·         The temperature dependence of the coercivities for the sintered samples can be discussed.

·         On page 3, line 110, in Bragg’s equation all the components used must be explained. What is “d”, n, lambda, etc. using the relevant work reported in the domain (doi.org/10.3390/nano7110356)?

·         The high angle shift and lattice constant decreases must be referred to in the literature such as 10.1016/j.electacta.2010.09.011; and 10.1016/j.jallcom.2008.12.143.

·         Was there any synergistic effect of using multicomponent elements for microstructure?

·         By improving the uniformity of the microstructure can the coercivity enhance?

·         In Figure 2, the red XRD curve must be labeled as Tb diffused.

·         The future development of this work can be given in section 4.

Author Response

Dear Reviewer, 

We thank the reviewers for the efforts given in reading our manuscript (materials-2330627) and having taken the time to provide valuable comments.  We have revised the manuscript according to your requirements carefully. Please see the attachment.

Round 2

Reviewer 2 Report

In this reviewer's opinion, the revised version is suitable for publication.